# PREDICT RESPONSIBLY: INCREASING FAIRNESS BY LEARNING TO DEFER

## ABSTRACT

When machine learning models are used for high-stakes decisions, they should predict accurately, fairly, and responsibly. To fulfill these three requirements, a model must be able to output a reject option (i.e. say "I Don't Know") when it is not qualified to make a prediction. In this work, we propose *learning to defer*, a method by which a model can defer judgment to a downstream decision-maker such as a human user. We show that learning to defer generalizes the rejection learning framework in two ways: by considering the effect of other agents in the decision-making process, and by allowing for optimization of complex objectives. We propose a learning algorithm which accounts for potential biases held by decision-makers later in a pipeline. Experiments on real-world datasets demonstrate that learning to defer can make a model not only more accurate but also less biased. Even when operated by highly biased users, we show that deferring models can still greatly improve the fairness of the entire pipeline.

## 1 INTRODUCTION

Recent machine learning advances have increased our reliance on learned automated systems in complex, high-stakes domains such as loan approvals (Burrell, 2016), medical diagnosis (Esteva et al., 2017), and criminal justice (Kirchner et al., 2016). This growing use of automated decision-making has raised questions about the obligations of classification systems. In many high-stakes situations, machine learning systems should satisfy (at least) three objectives: predict accurately (predictions should be broadly effective indicators of ground truth), predict fairly (predictions should be unbiased with respect to different types of input), and predict responsibly (predictions should not be made if the model cannot confidently satisfy the previous two objectives).

Given these requirements, we propose *learning to defer*. When deferring, the algorithm does not output a prediction; rather it says "I Don't Know" (IDK), indicating it has insufficient information to make a responsible prediction, and that a more qualified external decision-maker (DM) is required. For example, in medical diagnosis, the deferred cases would lead to more medical tests, and a second expert opinion. Learning to defer extends the common rejection learning framework (Chow, 1957; Cortes et al., 2016) in two ways. Firstly, it considers the expected output of the DM on each example, more accurately optimizing the output of the joint DM-model system. Furthermore, it can be used with a variety of training objectives, whereas most rejection learning research focuses solely on classification accuracy. We believe that algorithms that can defer, i.e., yield to more informed decision-makers when they cannot predict responsibly, are an essential component to accountable and reliable automated systems.

In this work, we show that the standard rejection learning paradigm (learning to *punt*) is inadequate, if these models are intended to work as part of a larger system. We propose an alternative decision making framework (learning to *defer*) to learn and evaluate these models. We find that embedding a deferring model in a pipeline can improve the accuracy and fairness of the *pipeline as a whole*, particularly if the model has insight into decision makers later in the pipeline. We simulate such a pipeline where our model can defer judgment to a better-informed decision maker, echoing real-world situations where downstream decision makers have more resources or information. We propose different formulations of these models along with a learning algorithm for training a model that can work optimally *with* such a decision-maker. Our experimental results on two real-world

datasets, from the legal and health domains, show that this algorithm learns models which, through deferring, can work with users to make fairer, more responsible decisions.

## 2 RELATED WORK

**Notions of Fairness.** One of the most challenging aspect of machine learning approaches to fairness is formulating an operational definition. Several works have focused on the goal of treating similar people similarly (individual fairness) and the resulting necessity of fair-awareness – showing that it may be essential to give the algorithm knowledge of the sensitive variable (Dwork et al., 2011; Zemel et al., 2013; Dwork et al., 2017).

Some definitions of fairness center around statistical parity (Kamiran & Calders, 2009; Kamishima et al., 2012), calibration (Pleiss et al., 2017; Guo et al., 2017) or disparate impact/equalized odds (Chouldechova, 2016; Hardt et al., 2016; Kleinberg et al., 2016; Zafar et al., 2017). It has been shown that disparate impact and calibration cannot be simultaneously satisfied (Chouldechova, 2016; Kleinberg et al., 2016). Hardt et al. (2016) present the related notion of "equal opportunity". In a subsequent paper, Woodworth et al. (2017) argue that in practice fairness criteria should be part of the learning algorithm, not post-hoc. Zafar et al. (2017) and Bechavod & Ligett (2017) develop and implement learning algorithms that integrate equalized odds into learning via regularization.

**Incorporating IDK.** While we are the first to propose learning to defer, some works have examined the "I don't know" (IDK) option (cf. *rejection learning*, see Cortes et al. (2016) for a thorough survey), beginning with Chow (1957; 1970) who studies the tradeoff between error-rate and rejection rate. Cortes et al. (2016) develop a framework for integrating IDK directly into learning. KWIK (Knows-What-It-Knows) learning is proposed in Li et al. (2011) as a theoretical framework. Attenberg et al. (2011) discuss the difficulty of a model learning what it doesn't know (particularly rare cases), and analyzes how human users can audit such models. Wang et al. (2017) propose a cascading model, which can be learned from end-to-end; higher levels can say IDK and pass the decision on to lower levels. Similarly, Kocak et al. (2017); Cortes et al. (2016); Bartlett & Wegkamp (2008) provide algorithms for saying IDK in classification and Ripley (2007) provides a statistical overview of the problem, including both "don't know" and "outlier" options. However, none of these works look at the fairness impact of this procedure.

A few papers have addressed topics related to both fairness and IDK. Bower et al. (2017) describe fair sequential decision making but do not have an IDK concept, nor do they provide a learning procedure. In Joseph et al. (2016), the authors show theoretical connections between KWIK-learning and a proposed method for fair bandit learning. Grgi-Hlaca et al. (2017) discuss fairness that can arise out of a mixture of classifiers. However, they do not provide a learning procedure, nor do they address sequential decision making, which we believe is of great practical importance. Varshney & Alemzadeh (2017) propose "safety reserves" and "safe fail" options which combine learning with rejection and fairness/safety, but do not suggest how such options may be learned or analyze the effect of a larger decision-making framework.

**AI Safety.** Finally, our work also touches on aspects of the AI safety literature - we provide a method by which a machine learns to work optimally with a human. This is conceptually similar to work such as Milli et al. (2017); Soares et al. (2015), which discuss the situations in which a robot should be compliant/cooperative with a human. The idea of a machine and human jointly producing a fair classifier also relates to Hadfield-Menell et al. (2016), which describes algorithms for machines to align with human values.

## 3 A JOINT DECISION-MAKING FRAMEWORK

Previous works in rejection learning (see Sec. 2) have proposed models that can choose to not classify (say IDK/reject). In these works, the standard method is to optimize the accuracy-IDK tradeoff: how much can a model improve its accuracy on the cases it *does* classify by saying IDK to some cases?

We find this paradigm inadequate. In many of the high-stakes applications this type of work is aimed at, an IDK is not the end of the story. Rather, a decision must be made eventually on every example, whether the model chooses to classify it or not.

Say our model is trained to detect melanoma, and when it says IDK, a human doctor can run an extra suite of medical tests. The model learns that it is very inaccurate at detecting amelanocytic (non-pigmented) melanoma, and says IDK if this might be the case. However, suppose that the doctor is even *less* accurate at detecting amelanocytic melanoma than the model is. Then, we may prefer the model to make a prediction despite its uncertainty. Conversely, if there are some patients that the doctor knows well, then they may have a more informed, nuanced opinion than the model. Then, we may prefer the model to say IDK more frequently relative to its internal uncertainty.

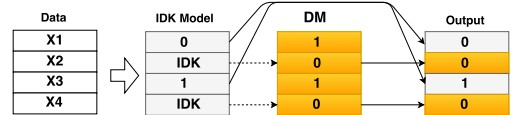

Figure 1: Data flow within larger system containing an IDK classifier (model). When the model predicts, the system outputs the model's prediction; when the model says IDK, the system outputs the decision-maker's (DM's) prediction. Rejection learning considers the "IDK Model" to be the system output.

Saying IDK on the wrong examples can also have fairness consequences. If the doctor's decisions bias against a certain group, then it is probably preferable for our model (if it is less biased) to defer less frequently on the cases of that group. In short, the model we train is part of a larger pipeline, and we should be training and evaluating the performance of *the pipeline with this model included*, rather than solely focusing on the model itself.

To enable this, we define a general two-step framework for decision-making (Fig 1). The first step is an automated model whose parameters we want to learn. The second step is some external decision maker (DM) which we do not have control over; this could be a human user or a more resource-intensive tool. The decision-making flow is done as a cascade, where the first-step model can either predict (positive/negative) or say IDK. If it predicts, the DM trusts the model completely, and outputs that prediction. However, if it says IDK, the DM makes its own decision. We assume that the DM is more powerful than the model we train — reflecting a number of practical scenarios where decision makers later in the chain have more resources for efficiency, security, or contextual reasons.

This system can be expressed by $Y^{sys} = sY^{DM} + (1-s)Y^{model}$, where $Y^{model}, Y^{DM}, Y^{sys}$ are the model, DM, and system output respectively, and $s$ is a binary IDK decision variable ($s = 1$ means IDK). Suppose we want to optimize some loss function $\mathcal{L}(Y, Y^{sys})$. In learning to defer, we train $s$ and $Y^{model}$ to optimize $\mathcal{L}(Y, sY^{DM} + (1-s)Y^{model})$. In Appendix A, we prove that learning to defer is equivalent to rejection learning for a broad class of loss functions, including classification error, if the DM is an oracle. Since our DM is rarely an oracle, learning to defer therefore yields a modeling advantage over rejection learning. Furthermore, learning to defer allows us to optimize a variety of objectives $\mathcal{L}(Y, Y^{sys})$, whereas most rejection learning research focuses on classification error.

In the rest of this work, we show how to learn fair IDK models in this framework. The paper proceeds as follows: in Sec. 4 we give some background on the fairness setup; in Sec. 5 we describe two methods of learning IDK models and how we may learn them in a fair way; and in Sec. 6 we give a learning algorithm for optimizing models to succeed in this framework. In Sec. 7 we show results on two real-world datasets.

## 4 BACKGROUND: FAIR CLASSIFICATION

In fair binary classification, we have data $X$, labels $Y$, predictions $\hat{Y}$, and sensitive attribute $A$, assuming for simplicity that $Y, \hat{Y}, A \in \{0, 1\}$. In this work we assume that $A$ is known (fair-aware) and that it is a single binary attribute (e.g., gender, race, age, etc.); extensions to more general settings are straightforward. The aim is twofold: firstly, that the classifier is *accurate* i.e., $Y_i = \hat{Y}_i$; and secondly, that the classifier is *fair with respect to A* i.e., $\hat{Y}$ does not discriminate unfairly against examples with a particular value of $A$. Classifiers with fairness constraints provably achieve worse error rates (cf. Chouldechova (2016); Kleinberg et al. (2016)). We thus define a loss function which trades off between these two objectives, relaxing the hard constraint proposed by models like (Hardt et al., 2016) and yielding a regularizer, similar to (Kamishima et al., 2012; Bechavod & Ligett, 2017). We use disparate impact (DI) as our fairness metric (Chouldechova, 2016), as it is becoming widely

used and also forms the legal basis for discrimination judgements in the U.S. Baldus & Cole (1980). Here we define a continuous relaxation of DI, using probabilistic output $p = P[Y = 1] \in [0, 1]$:

$$DI_{reg,Y=0}(Y, A, p) = |E(p|A = 0, Y = 0) - E(p|A = 1, Y = 0)|$$
$$DI_{reg,Y=1}(Y, A, p) = |E(1 - p|A = 0, Y = 1) - E(1 - p|A = 1, Y = 1)|$$
$$DI_{reg}(Y, A, p) = \frac{1}{2}(DI_{reg,Y=0}(Y, A, p) + DI_{reg,Y=1}(Y, A, p))$$

(1)

Note that constraining $DI = 0$ is equivalent to equalized odds (Hardt et al., 2016). If we constrain $p \in \{0, 1\}$, we say we are using *hard thresholds*; allowing $p \in [0, 1]$ is *soft thresholds*. We include a hyperparameter $\alpha$ to balance accuracy and fairness; there is no "correct" way to weight these. When we learn such a model, $p$ is a function of $X$ parametrized by $\theta$. Our regularized fair loss function ($\mathcal{L}_{Fair}$, or $\mathcal{L}_F$) combines cross-entropy for accuracy with this fairness metric:

$$\mathcal{L}_F(Y, A, X; \theta) = -\left[\sum_i Y_i \log p(x_i; \theta) + (1 - Y_i) \log(1 - p(x_i; \theta))\right] + \alpha DI_{reg}(Y, A, p(X; \theta))$$

(2)

## 5   SAYING IDK: LEARNING TO PUNT

We now discuss two model formulations that can output IDK: ordinal regression, and neural networks with weight uncertainty. Both of these models build on binary classifiers by allowing them to express some kind of uncertainty. In this section, we discuss these IDK models and how to train them to be fair; in the following section we address how to train them to take into account the downstream decision-maker.

### 5.1   ORDINAL REGRESSION

We extend binary classifiers to include a third option, yielding a model that can classify examples as "positive", "negative" or "IDK". This allows the model to *punt*, i.e., to output IDK when it prefers not to commit to a positive or negative prediction. We base our IDK models on ordinal regression with three categories (positive, IDK, and negative). These models involve learning two thresholds $\tau = (t_0, t_1)$ (see Figure 2). We can train with either hard or soft thresholds. If soft, each threshold $t_i, i \in \{0, 1\}$ is associated with a sigmoid function $\sigma_i$, where $\sigma_i(x) = \sigma(x - t_i)$; recall that $\sigma(x) = \frac{1}{1+e^{-x}}$.

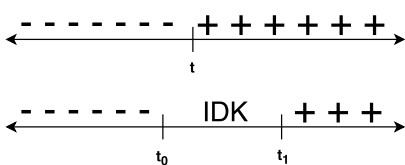

Figure 2: Binary classification (one threshold) vs. IDK classification (two thresholds)

.

These thresholds yield an ordinal regression, which produces three values for every score $x$: $P(x), I(x), N(x) \in [0, 1]$, s.t. $P(x) + I(x) + N(x) = 1$. Using hard thresholds simply restricts $P, I, N \in \{0, 1\}^3$ (one-hot vector). We can also calculate a score $p(x) \in [0, 1]$, which we interpret as the model's prediction disregarding uncertainty. These values are:

$$P(x) = \sigma_1(x); \quad I(x) = \sigma_0(x) - \sigma_1(x); \quad N(x) = 1 - \sigma_0(x); \quad p(x) = \frac{P(x)}{P(x) + N(x)} \quad (3)$$

$P$ represents the model's bet that $x$ is "positive", $N$ the bet that $x$ is "negative", and $I$ is the model hedging its bets; this rises with uncertainty. Note that $p$ is minimized at $P = N$; this is also where $I$ is maximized. At test time, we use the thresholds to partition the examples. On each example, the model outputs a score $x \in \mathcal{R}$ (the logit for the ordinal regression), and a prediction $p$. If $t_0 < x < t_1$, then we replace the model's prediction $p$ with IDK. If $x \leq t_0$ or $x \geq t_1$, we leave $p$ as is. To encourage fairness, we can learn a separate set of thresholds for each group: $(t_{0,A=0}, t_{1,A=0})$ and $(t_{0,A=1}, t_{1,A=1})$; then apply the appropriate set of thresholds to each example. We can also regularize IDK classifiers for fairness. When training this model, $P$, $I$, and $N$ are parametrized functions, with model parameters $\theta$ and thresholds $\tau$. Using soft thresholds, the regularized loss

function $\mathcal{L}_{FairPunt}$ ($\mathcal{L}_{FP}$) is:

$$\mathcal{L}_{FP}(Y, A, X; \theta, \tau) = - \left[ \sum_i Y_i \log P(x_i; \theta, \tau) + (1 - Y_i) \log N(x_i; \theta, \tau) - \gamma \log I(x_i; \theta, \tau) \right]$$
$$+ \alpha(DI_{reg}(Y, A, P(X; \theta, \tau)) + DI_{reg}(Y, A, N(X; \theta, \tau)))$$
(4)

Note that we add a term penalizing $I(X)$, to prevent the trivial solution of always outputting IDK. In addition, we regularize the disparate impact for $P(X)$ and $N(X)$ separately. This was not necessary in the binary case, since these two probabilities would have always summed to 1. We learn soft thresholds end-to-end; for hard thresholds we use a post-hoc thresholding scheme (see Appendix D).

## 5.2 BAYESIAN WEIGHT UNCERTAINTY

We can also take a Bayesian approach to uncertainty by learning a distribution over the weights of a neural network (Blundell et al., 2015). In this method, we use variational inference to approximate the posterior distribution of the model weights given the data. When sampling from this distribution, we can obtain an estimate of the uncertainty. If sampling several times yields widely varying results, we can state the model is uncertain on that example.

This model outputs a prediction $p$ and an uncertainty $\pi$ for example $x$. We calculate these by sampling $J$ times from the model, yielding $J$ predictions $z_j \in [0, 1]$. Our prediction $p$ is the sample mean $\mu = \frac{1}{J} \sum_{j=1}^{J} z_j$. To numerically represent our uncertainty, we can use signal-to-noise ratio, defined as $S = \frac{|\mu - 0.5|}{\sigma}$, based on $\mu$ and the sample standard deviation $\sigma = \sqrt{\frac{\sum_{j=1}^{J} (z_j - \mu)^2}{J - 1}}$. The reciprocal of this ($\pi = 1/S$) allows high values to be more uncertain, while $\pi = \sigma(\log(1/S))$ (where $\sigma$ is the logistic function) yields uncertainty values in a $[0, 1]$ range. At test time, the system can threshold this uncertainty; any example with uncertainty beyond a threshold is punted to the DM.

We can regularize this Bayesian model to improve fairness as in the standard binary classifier. In the likelihood term for the variational inference, we can simply add the disparate impact regularizer (Eq. 1), making solutions of low disparate impact more likely. With weights $w$ and variational parameters $\theta$, our variational lower bound $\mathcal{L}_{\mathcal{V}}$ is then:

$$\mathcal{L}_{\mathcal{V}}(Y, A, X, w; \theta) = -KL[q(w|\theta)||Prior(w)] +$$
$$\mathbb{E}_{q(w|\theta)} \left[ - \left[ \sum_i Y_i \log p(x_i; \theta) + (1 - Y_i) \log (1 - p(x_i; \theta)) \right] + \alpha DI_{reg}(Y, A, p(X; \theta)) \right]$$
(5)

## 6 LEARNING TO DEFER

IDK models come with a consequence: when a model punts, the prediction is made instead by some external, possibly biased decision maker (DM) e.g., a human expert. In this work we assume that this DM is possibly biased, but is more accurate than the model; perhaps the DM is a judge with detailed information on repeat offenders, and with more information about the defendant than the model has, or a doctor who can conduct a suite of complex medical tests.

Here we introduce a distinction between *learning to punt* and *learning to defer*. In learning to punt, the goal is absolute: the model's aim is to identify the examples where it has a low chance of being correct. In learning to defer, the model has some information about the DM and takes this into account in its IDK decisions. Hence the goal is relative: the model's aim is to identify the examples where the DM's chance of being correct is much higher than its own. If the model punts mostly on cases where the DM is very inaccurate or unfair, then the joint predictions made by the model-DM pair may be poor, even if the model's own predictions are good. We can think of learning to punt as *DM-unaware learning*, and learning to defer as *DM-aware learning*.

To conduct DM-aware learning, we can modify the model presented in Section 5 to take an extra input: the DM's scores on every case in the training set. The model is then optimized for some loss function $\mathcal{L}(Y, A, X)$; for our purposes, this loss will be a combination of accuracy and fairness. We propose the following general method, drawing inspiration from mixture-of-experts (Jacobs et al., 1991). We introduce a mixing parameter $\pi$ for each example $x$, which is the *probability of deferral*; that is, the probability that the DM makes the final decision on the example $x$, rather than the model. Then, $1 - \pi$ is the probability that the model's decision becomes the final output of the system. Let $s \sim Ber(\pi)$. Our mixing parameter $\pi$ corresponds to our model's uncertainty estimate — $I(x)$ in ordinal regression, $\sigma(\log(\frac{1}{S}))$ in the Bayesian neural network. Let $p$ be the first stage model's predictions and $\tilde{Y}$ be the DM's predictions. We can express the joint system's predictions $\hat{p}$ as

$$\hat{p} = s\tilde{Y} + (1 - s)p; \quad s \in \{0, 1\}; \hat{p}, \tilde{Y}, p \in [0, 1] \tag{6}$$

In learning this model, we can parametrize $p$ and $\pi$ by $\theta = (\theta_p, \theta_\pi)$, which may be shared parameters. We can then define our loss function ($\mathcal{L}_{Defer}$, or $\mathcal{L}_D$) as an expectation over the Bernoulli variables $s_i \sim Ber(\pi(x_i, \theta_\pi))$:

$$
\begin{aligned}
\mathcal{L}_D(Y, A, X; \theta) =& \mathbb{E}_s \mathcal{L}(Y, A, X; \theta) \\
=& \mathbb{E}_s \Bigg[ -\sum_i \Big[ Y_i \log \hat{p}(x_i; \theta) + (1 - Y_i) \log(1 - \hat{p}(x_i; \theta)) - \gamma \log \pi(x_i; \theta_\pi) \Big] \\
& + \alpha DI_{reg}(Y, A, X; \theta) \Bigg]
\end{aligned} \tag{7}
$$

We call this *learning to defer*. When optimizing this function, the model learns to recognize when there is relevant information that is *not* present in the data it has been given by comparing its own predictions to the DM's predictions. Full details of how this loss function is calculated are provided in Appendix E.

## 7 RESULTS

**Experimental Setup**. To evaluate our models, we measure three quantities: classification error, disparate impact, and deferral rate. We train an independent model to simulate predictions made by an external DM. This DM is trained on a version of the dataset with extra attributes, simulating the extra knowledge/accuracy that the DM may have. However, the DM is *not* trained to be fair. When our model outputs IDK we take the output of the DM instead (see Figure 1).

**Datasets and Experiment Details**. We show results on two datasets: the COMPAS dataset (Kirchner et al., 2016), where we predict a defendant's recidivism (committing a crime while on bail) without discriminating by race, and the Heritage Health dataset, where we predict a patient's Charlson Index (a comorbidity indicator related to likelihood of death) without discriminating by age. For COMPAS, we give the DM the ground truth for a defendant's violent recidivism; for Health, we give the DM the patient's primary condition group. Appendix C contains additional details on both datasets.

We trained all models using a one-hidden-layer fully connected neural network with a logistic or ordinal regression on the output, where appropriate. We used 5 sigmoid hidden units for COMPAS and 20 sigmoid hidden units for Health. We used ADAM (Kingma & Ba, 2014) for gradient descent. We split the training data into 80% training, 20% validation, and stopped training after 50 consecutive epochs without achieving a new minimum loss on the validation set.

In the ordinal regression model, we trained with soft thresholds since we needed the model to be differentiable end to end. In the post-hoc model, we searched threshold space in a manner which did not require differentiability, so we used hard thresholds. This is equivalent to an ordinal regression which produces one-hot vectors i.e. $P, I, N \in \{0, 1\}$. See Appendices D and E for additional details on both of these cases.

**Displaying Results**. Each model contains hyperparameters, such as the coefficients $(\alpha, \gamma)$ for training and/or post-hoc optimization. We show the results of several models, with various hyperparameter settings, to illustrate how they mediate the tradeoff of accuracy and fairness. Each plotted point is a median of 5 runs at a given hyperparameter setting. We only show points on the Pareto front of

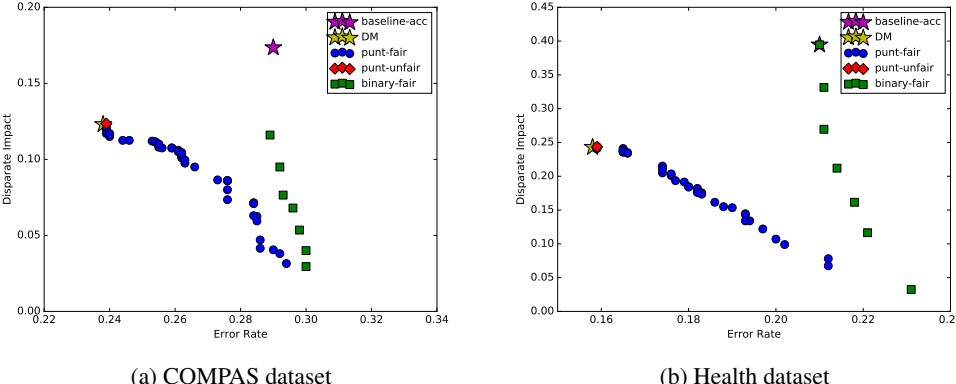

(a) COMPAS dataset           (b) Health dataset

Figure 3: Comparing the performance of punting IDK and binary models, with and without a fairness regularizer. The figure illustrates the trade-off between accuracy (x-axis) and fairness (y-axis). Bottom left hand corner is optimal. The purple star is a baseline model, trained only to optimize accuracy; green squares is a model also optimizing fairness; the red diamond optimizes accuracy while allowing IDK; and blue circles are the full model with all three terms. Yellow star shows the second stage model DM alone. Each point is the median of 5 runs on the test set at a given hyperparameter setting.

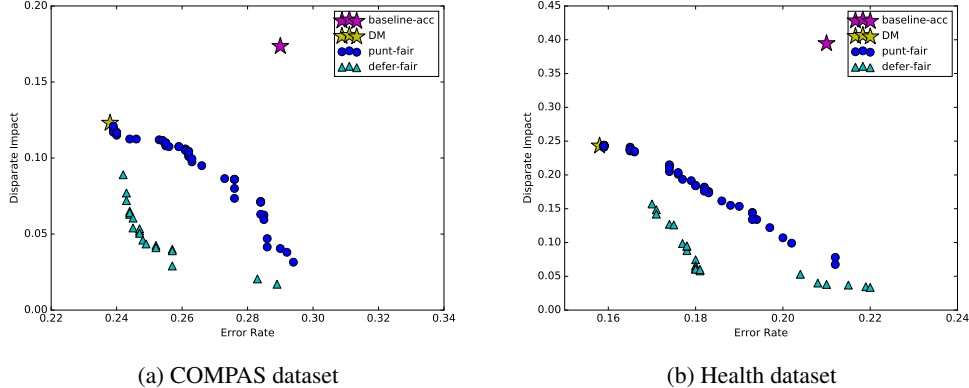

(a) COMPAS dataset           (b) Health dataset

Figure 4: Comparison of DM-aware and -unaware learning (i.e., Section 5 vs. Section 6). Most of the results are the same as in Figure 3; the new results here show the performance of a DM-aware model (defer-fair), depicted by the green triangles. Of particular note is the improvement of this model relative to the punting model (the blue circles).

the results, i.e., those for which no other point had both better error and DI. Finally, all results are calculated on a held-out test set.

## 7.1 LEARNING TO PUNT AND DEFER

In Figure 3, we compare punting models to binary models, with and without fairness regularization. These IDK models have *not* learned to defer, i.e. they did not receive access to the DM scores during training (see Fig. 4). The results show that, on both datasets, the IDK models achieve a stronger combination of fairness and accuracy than the binary models. Graphically, we observe this by noting that the line of points representing IDK model results are closer to the lower left hand corner of the plot than the line of points representing binary model results. Some of this improvement is driven by the extra accuracy in the DM. However, we note that the model-DM combination achieves a more effective accuracy-fairness tradeoff than any of the three baselines: the accurate but unfair DM; the fair but inaccurate binary model with DI regularization; and the unfair and inaccurate unregularized

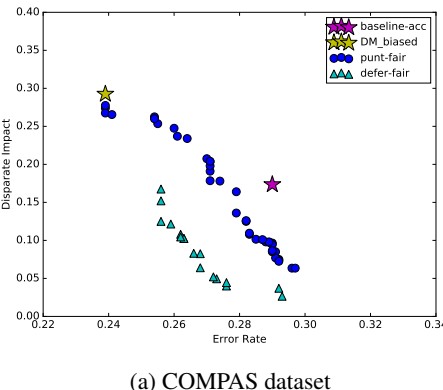

(a) COMPAS dataset

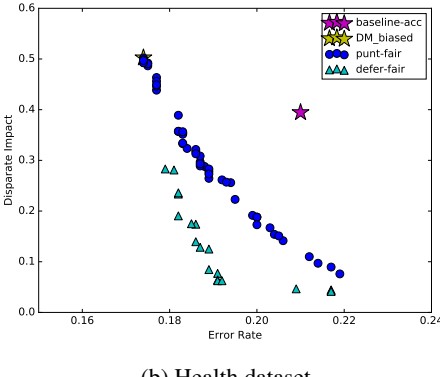

(b) Health dataset

Figure 5: Results obtained with a highly biased DM (trained with $\alpha = -0.1$). Note that DM-aware still improves relative to DM-unaware learning in this case.

binary model. Learning to punt can therefore be a valuable tool for anyone who designs or oversees a many-part system - a simple first stage capable of expressing uncertainty can improve the fairness of a more accurate DM.

Figure 4 demonstrates a clear improvement over the punting models (DM-unaware, Sec. 5). If we have access to examples of past DM behavior, learning to defer provides an effective way to improve the fairness of the *entire system*. For insight here, we can inspect the different roles IDK plays in their respective loss functions. In the DM-unaware IDK model, punting is penalized at a constant rate, determined by $\gamma$. However, in the DM-aware model, deferring penalized in a way which is *dependent on the output of the DM* on that example. We can consider the unaware model to be optimizing the expected DM-aware loss function for a DM with constant expected loss on each examples, such as an oracle (see Appendix A). Then, we can see any improvement by the DM-aware model as effective identification of the examples on which the expected loss of the DM is unusually high; in other words, identifying the inconsistencies or biases of the DM.

## 7.2 RESULTS: DEFERRING TO A BIASED DM

One advantage of deferring is that it can account for specific characteristics of a DM. To test this, we considered the case of a DM which is extremely biased (Fig. 5). We find that the advantage of a deferring model holds in this case, as it compensates for the DM's extreme bias. We can further analyze where the model chooses to defer. Recall that the DM is given extra information; in this case the violent recidivism of the defendant (true for about 7% of the dataset), which is difficult to predict from the other attributes. Fig. 6 compares the IDKs predicted by a punting model and a deferring model - split by group (race) on the left, and by group and violent recidivism on the right. Both models achieved roughly 27% error; the deferring model had 2% DI and the punting model had 4%. On the left, we see that the deferring model says IDK to more black people (the pink bar). On the right however, we see that the de-



Figure 6: Comparison of IDK predictions between deferring and punting model on COMPAS dataset. Total IDKs are normalized to 1. A = 1 is the protected group (black people). Y is the additional information given to the DM (violent recidivism) - Y = 1 means violent recidivism occurred. Pink is A = 1, green is A = 0. Diagonal cross-hatch is Y = 0, horizontal cross-hatch is Y=1. DM is trained to be extremely unfair with $\alpha = -0.1$.

ferring model says IDK to a higher percentage of violently recidivating non-black people, and a lower percentage of violently recidivating black people. This improves DI - the extra information the DM has received is more fully used on the non-protected group. The punting model cannot

adjust this way; the deferring model can, since it receives noisy access to this information through the DM scores in training.

## 8 CONCLUSION

In this work, we propose the idea of learning to defer. We propose a model which learns to defer fairly, and show that these models can better navigate the accuracy-fairness tradeoff. We also consider deferring models as one part of a decision pipeline. To this end, we provide a framework for evaluating deferring models by incorporating other decision makers' output into learning. We give an algorithm for learning to defer in the context of a larger system, and show how to train a deferring model to optimize the performance of the pipeline as a whole.

This is a powerful, general framework, with ramifications for many complex domains where automated models interact with other decision-making agents. A model with a low deferral rate could be used to cull a large pool of examples, with all deferrals requiring further examination. Conversely, a model with a high deferral rate can be thought of as flagging the most troublesome, incorrect, or biased decisions by a DM, with all non-deferrals requiring further investigation. Automated models often operate within larger systems, with many moving parts. Through deferring, we show how models can learn to predict responsibly within their surrounding systems. Automated models often operate within larger systems, with many moving parts. Through deferring, we show how models can learn to predict responsibly within their surrounding systems. Building models which can defer to more capable decision makers is an essential step towards fairer, more responsible machine learning.

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

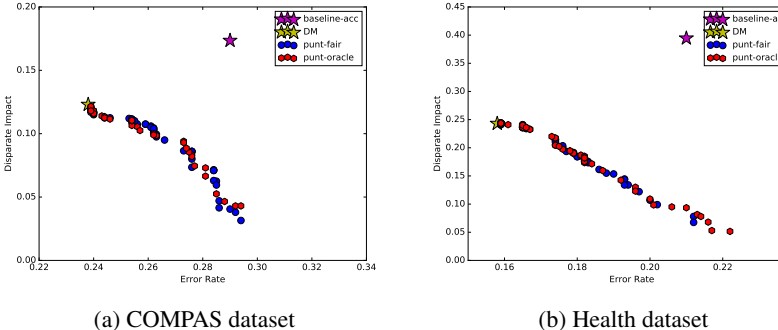

(a) COMPAS dataset                    (b) Health dataset

Figure 7: Comparing model performance between expected loss training with oracle as DM to IDK training unaware of DM. At test time, same DM is used.

Smitha Milli, Dylan Hadfield-Menell, Anca Dragan, and Stuart Russell. Should Robots be Obedient? *arXiv:1705.09990 [cs]*, May 2017. URL http://arxiv.org/abs/1705.09990. arXiv: 1705.09990.

Geoff Pleiss, Manish Raghavan, Felix Wu, Jon Kleinberg, and Kilian Q. Weinberger. On Fairness and Calibration. *arXiv:1709.02012 [cs, stat]*, September 2017. URL http://arxiv.org/abs/1709.02012. arXiv: 1709.02012.

Brian D Ripley. *Pattern recognition and neural networks*. Cambridge university press, 2007.

Nate Soares, Benja Fallenstein, Stuart Armstrong, and Eliezer Yudkowsky. Corrigibility. In *Workshops at the Twenty-Ninth AAAI Conference on Artificial Intelligence*, 2015.

Kush R Varshney and Homa Alemzadeh. On the safety of machine learning: Cyber-physical systems, decision sciences, and data products. *Big data*, 5(3):246–255, 2017.

Xin Wang, Yujia Luo, Daniel Crankshaw, Alexey Tumanov, and Joseph E. Gonzalez. IDK Cascades: Fast Deep Learning by Learning not to Overthink. *arXiv:1706.00885 [cs]*, June 2017. URL http://arxiv.org/abs/1706.00885. arXiv: 1706.00885.

Blake Woodworth, Suriya Gunasekar, Mesrob I. Ohannessian, and Nathan Srebro. Learning Non-Discriminatory Predictors. *arXiv:1702.06081 [cs]*, February 2017. URL http://arxiv.org/abs/1702.06081. arXiv: 1702.06081.

Muhammad Bilal Zafar, Isabel Valera, Manuel Gomez Rodriguez, and Krishna P. Gummadi. Fairness Beyond Disparate Treatment & Disparate Impact: Learning Classification without Disparate Mistreatment. *arXiv:1610.08452 [cs, stat]*, pp. 1171–1180, 2017. doi: 10.1145/3038912.3052660. URL http://arxiv.org/abs/1610.08452. arXiv: 1610.08452.

Richard Zemel, Yu Wu, Kevin Swersky, Toni Pitassi, and Cynthia Dwork. Learning Fair Representations. In *PMLR*, pp. 325–333, February 2013. URL http://proceedings.mlr.press/v28/zemel13.html.

## A  COMPARISON OF ORACLE TRAINING TO IDK DM-UNAWARE

In Section 7.1, we discuss that DM-unaware IDK training is similar to DM-aware training, except with a training DM who treats all examples similarly, in some sense. Here we show experimental evidence. The plots in Figure 7 compare these two models: DM-unaware, and DM-aware with an oracle at training time, and the standard DM at test time. We can see that these models trade off between error rate and DI in almost an identical manner.

We can show that when for a broad class of objective functions (including classification error and cross entropy), these are provably equivalent. Note that this class does not include our fairness regularizer; for that we show the experimental evidence in Figure 7.

Let $Y$ be the ground truth label, $Y^{DM}$ be the DM output, $Y^{model}$ be the model output, $s$ be the IDK decision indicator ($s = 0$ for predict, $s = 1$ for IDK), and $Y^{sys}$ be the output of the joint DM-model system. Suppose these are all binary variables.

The standard rejection learning (DM-unaware) loss (which has no concept of $Y^{DM}$) is (Cortes et al., 2016):

$$\mathcal{L}_{punt}(Y, Y^{model}, s) = \sum_i (1 - s_i)\mathbb{1}[Y_i \neq Y_i^{model}] + s\gamma_{punt} \tag{8}$$

We can describe the learning-to-defer system output as

$$Y^{sys} = sY^{DM} + (1 - s)Y^{model} \tag{9}$$

If we wish to train the system output $Y^{sys}$ to optimize some loss function $\mathcal{L}(Y, Y^{sys})$, we can simply train $s$ and $Y^{model}$ to optimize $\mathcal{L}(Y, sY^{DM} + (1 - s)Y^{model})$. This deferring framework is strictly more expressive than the rejection learning model, as it can be used on many different objectives $\mathcal{L}$, while rejection learning is mostly used with classification accuracy. We now show that if we take the DM to be an oracle (always outputs ground truth), learning to defer reduces to rejection learning for a broad class of objective functions, including classification error, cross entropy, and mean squared error.

**Theorem.** *Let $Y^{sys} = sY^{DM} + (1-s)Y^{model}$, where $s \in \{0, 1\}$. Let $\mathcal{L}(Y, Y^{sys}) = \sum_i \ell(Y_i, Y_i^{sys})$ be the objective we aim to minimize, where $Y_i = \arg\min_{Y_i^{sys}} \ell(Y_i, Y_i^{sys})$ and $\ell(Y_i, Y_i) = 0$. Then, if the DM is an oracle, the learning-to-defer and learning-to-punt objectives are equivalent.*

**Proof.** As in Eq. 8, the standard rejection learning objective is

$$\mathcal{L}_{punt}(Y, Y^{model}, s) = \sum_i (1 - s_i)\ell(Y_i, Y_i^{model}) + s\gamma_{punt} \tag{10}$$

where the first term encourages a low loss $\ell$ for non-IDK examples and the second term penalizes IDK at a constant rate, with $\gamma_{punt} \geq 0$. In rejection learning, $\ell$ is usually classification error (cf. Cortes et al. (2016); Chow (1957)). Note that this objective has no notion of DM output ($Y^{DM}$).

If we include a similar $\gamma_{defer}$ penalty, the deferring loss function is

$$\begin{aligned}
\mathcal{L}_{defer}(Y, Y^{DM}, Y^{model}, s) &= \sum_i \ell(Y_i, Y_i^{sys}) + s\gamma_{defer} \\
&= \sum_i \ell(Y_i, s_iY_i^{DM} + (1 - s_i)Y_i^{model}) + s\gamma_{defer} \\
&= \sum_i s_i\ell(Y_i, Y_i^{DM}) + (1 - s_i)\ell(Y_i, Y_i^{model}) + s\gamma_{defer}
\end{aligned} \tag{11}$$

Now, if the DM is an oracle, then $Y^{DM} = Y$, meaning $\ell(Y, Y^{DM}) = 0$, giving us

$$\begin{aligned}
\mathcal{L}_{defer}(Y, Y^{DM}, Y^{model}, s) &= \sum_i s_i \cdot 0 + (1 - s_i)\ell(Y_i, Y_i^{model}) + s\gamma_{defer} \\
&= \sum_i (1 - s_i)\ell(Y_i, Y_i^{model}) + s\gamma_{defer} \\
&= \mathcal{L}_{punt}(Y, Y^{model}, s)
\end{aligned} \tag{12}$$

if we set $\gamma_{defer} = \gamma_{punt}$. ∎

## B    RESULTS: BINARY CLASSIFICATION WITH FAIR REGULARIZATION

The results in Figures 8 and 9 roughly replicate the results from (Bechavod & Ligett, 2017), who also test on the COMPAS dataset. Their results are slightly different for two reasons: 1) we use a 1-layer NN and they use logistic regression; and 2) our training/test splits are different from theirs - we have more examples in our training set. However, the main takeaway is similar: regularization with a disparate impact term is a good way to reduce DI without making too many more errors.

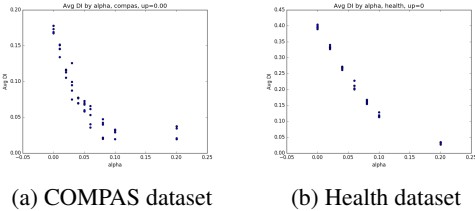

(a) COMPAS dataset        (b) Health dataset

Figure 8: Relationship of DI to $\alpha$, the coefficient on the DI regularizer, 5 runs for each value of $\alpha$

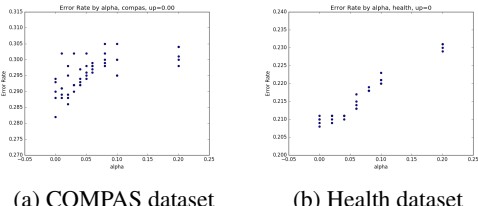

(a) COMPAS dataset        (b) Health dataset

Figure 9: Relationship of error rate to $\alpha$, the coefficient on the DI regularizer, 5 runs for each value of $\alpha$

## C    DATASET DETAILS

We show results on two datasets. The first is the COMPAS recidivism dataset, made available by ProPublica (Kirchner et al., 2016) [1]. This dataset concerns recidivism: whether or not a criminal defendant will commit a crime while on bail. The goal is to predict whether or not the person will recidivate, and the sensitive variable is race (split into black and non-black). We used information about counts of prior charges, charge degree, sex, age, and charge type (e.g., robbery, drug possession). We provide one extra bit of information to our DM - whether or not the defendant *violently* recidivated. This clearly delineates between two groups in the data - one where the DM knows the correct answer (those who violently recidivated) and one where the DM has no extra information (those who did not recidivate, and those who recidivated non-violently). This simulates a real-world scenario where a DM, unbeknownst to the model, may have extra information on a subset of the data. The simulated DM had a 24% error rate, better than the baseline model's 29% error rate. We split the dataset into 7718 training examples and 3309 test examples.

The second dataset is the Heritage Health dataset [2]. This dataset concerns health and hospitalization, particularly with respect to insurance. For this dataset, we chose the goal of predicting the Charlson Index, a comorbidity indicator, related to someone's chances of death in the next several years. We binarize the Charlson Index of a patient as 0/greater than 0. We take the sensitive variable to be age and binarize by over/under 70 years old. This dataset contains information on sex, age, lab test, prescription, and claim details. The extra information available to the DM is the primary condition group of the patient (given in the form of a code e.g., 'SEIZURE', 'STROKE', 'PNEUM'). Again, this simulates the situation where a DM may have extra information on the patient's health that the algorithm does not have access to. The simulated DM had a 16% error rate, better than the baseline model's 21% error rate. We split the dataset into 46769 training examples and 20044 test examples.

## D    DETAILS ON OPTIMIZATION: HARD THRESHOLDS

We now explain the post-hoc threshold optimization search procedure we used. In theory, any procedure can work. Since it is a very small space (one dimension per threshold = 4 dimensions), we used a random search. We sampled 1000 combinations of thresholds, picked the thresholds which minimized the loss on one half of the test set, and evaluated these thresholds on the other half

---

[1] downloaded from https://github.com/propublica/compas-analysis

[2] Downloaded from https://www.kaggle.com/c/hhp

of the test set. We do this for several values of $\alpha, \gamma$ in thresholding, as well as several values of $\alpha$ for the original binary model.

We did not sample thresholds from the [0, 1] interval uniformly. Rather we used the following procedure. We sampled our lower thresholds from the scores in the training set which were below 0.5, and our upper thresholds from the scores in the training set which were above 0.5. Our sampling scheme was guided by two principles: this forced 0.5 to always be in the IDK region; and this allowed us to sample more thresholds where the scores were more dense. If only choosing one threshold per class, we sampled from the entire training set distribution, without dividing into above 0.5 and below 0.5.

This random search was significantly faster than grid search, and no less effective. It was also faster and more effective than gradient-based optimization methods for thresholds - the loss landscape seemed to have many local minima.

## E    DETAILS ON TRAINING WITH EXPECTED LOSS: SOFT THRESHOLDS

We go into more detail regarding the regularization term for expected disparate impact in Equation 7. When using soft thresholds, it is not trivial to calculate the expected disparate impact regularizer:

$$
\begin{aligned}
DI_{exp}(Y, A, \hat{Y}) &= \mathbb{E}_{\hat{\pi}} DI_{reg}(Y, A, \hat{Y}) \\
&= \mathbb{E}_{\hat{\pi}} \frac{1}{2} (DI_{reg,Y=0}(Y, A, p) + DI_{reg,Y=1}(Y, A, p))
\end{aligned}
\tag{13}
$$

due to the difficulties involved in taking the expected value of an absolute value. We instead chose to calculate a version of the regularizer with squared underlying terms:

$$
\begin{aligned}
DI_{soft}(Y, A, \hat{Y}) &= \mathbb{E}_{\hat{\pi}} \frac{1}{2} (DI_{reg,Y=0}(Y, A, \hat{Y})^2 + DI_{reg,Y=1}(Y, A, \hat{Y})^2) \\
&= \mathbb{E}_{\hat{\pi}} (E(\hat{Y}_i | A = 0, Y = 0) - E(\hat{Y}_i | A = 1, Y = 0))^2 \\
&\quad + \mathbb{E}_{\hat{\pi}} (E(1 - \hat{Y}_i | A = 0, Y = 1) - E(1 - \hat{Y}_i | A = 1, Y = 1))^2 \\
&= \mathbb{E}_{\hat{\pi}} \left( \frac{\sum_i^n (1 - Y_i)(1 - A_i)\hat{Y}_i}{\sum_i^n (1 - Y_i)(1 - A_i)} - \frac{\sum_i^n (1 - Y_i)A_i\hat{Y}_i}{\sum_i^n (1 - Y_i)A_i} \right)^2 \\
&\quad + \mathbb{E}_{\hat{\pi}} \left( \frac{\sum_i^n Y_i(1 - A_i)(1 - \hat{Y}_i)}{\sum_i^n Y_i(1 - A_i)} - \frac{\sum_i^n Y_i A_i(1 - \hat{Y}_i)}{\sum_i^n Y_i A_i} \right)^2
\end{aligned}
\tag{14}
$$

Then, we can expand $\hat{Y}_i$ as

$$
\hat{Y}_i = \hat{\pi}_i \tilde{Y}_i + (1 - \hat{\pi}_i) p_i
\tag{15}
$$

where $\tilde{Y}_i \in [0, 1]$ and $S(x_i) \in [0, 1]$ are the DM and machine predictions respectively. For brevity we will not show the rest of the calculation, but with some algebra we can obtain a closed form expression for $DI_{soft}(Y, A, \hat{Y})$ in terms of $Y, A, \tilde{Y}$ and $S$.

## F    RESULTS: LEARNING TO DEFER, BY DEFERRAL RATE

Models which rarely defer behave very differently from those which frequently defer. In Figure 10, we break down the results from Section 7.1 by deferral (or punting) rate. First, we note that even for models with similar deferral rates, we see a similar fairness/accuracy win for the DM-aware models. Next, we can look separately at the low and high deferral rate models. We note that the benefit of DM-aware training is much larger for high deferral rate models. This suggests that the largest benefit of learning to defer comes from a win in fairness, rather than accuracy.

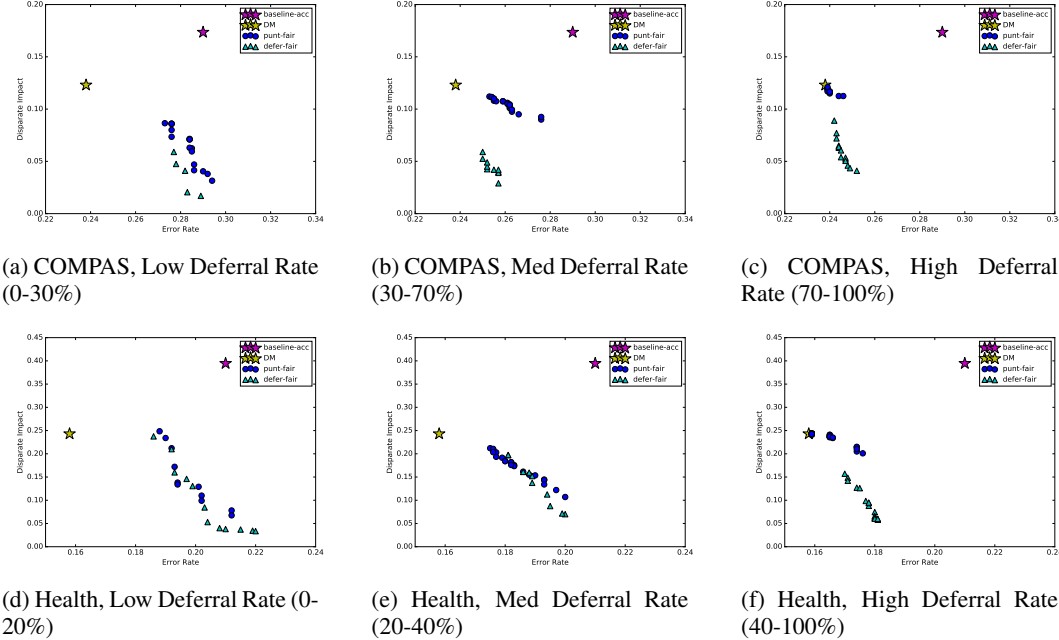

(a) COMPAS, Low Deferral Rate (0-30%)

(b) COMPAS, Med Deferral Rate (30-70%)

(c) COMPAS, High Deferral Rate (70-100%)

(d) Health, Low Deferral Rate (0-20%)

(e) Health, Med Deferral Rate (20-40%)

(f) Health, High Deferral Rate (40-100%)

Figure 10: Comparison of DM-aware and -unaware learning. Split into 3 bins, low, medium, and high deferral rate for each dataset. Bins are different between datasets due to the differing distributions of deferral rate observed during hyperparameter search.

