# OpenReview forum: "Predict Responsibly: Increasing Fairness by Learning to Defer"
_ICLR.cc/2018/Conference — Invite to Workshop Track_

### Official Review · AnonReviewer3 · 2017-11-27
**not in scope of call for papers**

**Rating:** 4
**Confidence:** 5

**Review:**

I like the direction this paper is going in by combining fairness objectives with deferment criteria and learning.  However, I do not believe that this paper is in the scope defined by the ICLR call for papers, as there is nothing related to learned representations in it.

I find it quite interesting that the authors go beyond 'classification with a reject option' to learning to defer, based on output predictions of the second decision maker.  However, the authors do not make this aspect of the work very clear until Section 6.  The distinction and contribution of this part is not made obvious in the abstract and introduction.  And it is not stated clearly whether there is any related prior work on this aspect.  I'm not sure there is, and if there isn't, then the authors should highlight that fact.  The authors should discuss more extensively how the real-world situation will play out in terms of having two different training labels per sample (one from the purported DM and another used for the training of the main supervised learning portion).  Typically it is DM labels that are the only thing available for training and what cause the introduction of unfairness to begin with.

Throughout the paper, it is not obvious or justified why certain choices are made, e.g. cross-entropy.

In the related work section, specifically Incorporating IDK, it would be good to discuss the work of Wegkamp et al., including the paper http://www.jmlr.org/papers/volume9/bartlett08a/bartlett08a.pdf and its loss function. Also, the work of https://doi.org/10.1145/2700832 would be nice to briefly discuss.

Also in the related work section, specifically AI Safety, it would be good to discuss the work of Varshney and Alemzadeh (https://doi.org/10.1089/big.2016.0051) --- in particular the 'safety reserves' and 'safe fail' sections which specifically address reject options and fairness respectively.

The math and empirical results all seem to be correct and interesting.

---

> ### Author Response · Authors · 2017-12-21
> **Response to Reviewer 3**
>
> Thank you for the review. We are glad you found the paper enjoyable and interesting. A couple of clarifying responses:
>
> -"I do not believe that this paper is in the scope defined by the ICLR call for papers, as there is nothing related to learned representations in it."
>
> In fact our paper does contains learned representations - our neural networks have a hidden layer. We have clarified this in the paper (by changing the phrase "one-layer" to "one-hidden-layer" i.e. one hidden layer of non-linear units and two layers of weights); we apologize for the miscommunication.
>
> And, we disagree about the out-of-scope critique.  The scope of ICLR is far broader than papers directly concerning the specifics of learned representations. A quick glance at the papers chosen for oral presentations at ICLR 2017 reveals papers on optimization, generalization, and privacy. Fairness and learning with rejection are accepted, important, and popular topics with a rich literature in the machine learning and deep learning communities; they are well within the scope of this conference.
>
> -"the authors do not make this aspect of the work very clear until Section 6. The distinction and contribution of this part is not made obvious in the abstract and introduction."
>
> We have edited the introduction and abstract to clarify our contribution.
>
> -"it is not stated clearly whether there is any related prior work on this aspect.  I'm not sure there is, and if there isn't, then the authors should highlight that fact."
>
> We have edited the Related Work and Introduction to clarify our contribution.
>
> -"The authors should discuss more extensively how the real-world situation will play out in terms of having two different training labels per sample.  Typically it is DM labels that are the only thing available for training and what cause the introduction of unfairness to begin with."
>
> Selective label bias is certainly a prominent problem in constructing datasets.  However, a decision-maker can display significant bias above and beyond selective label bias, so our method can still be very useful. For instance, all models that train on the COMPAS dataset (a standard dataset in fairness research), by necessity only know the ground truth for defendants who received bail; we can never know if those who did not receive bail would have recidivated. Yet many papers publish with results on this dataset, and show useful results. This is an issue larger than our work, encompassing the field of machine learning as a whole - every paper must consider the effects of bias in the data generation process.
>
> -"it is not obvious or justified why certain choices are made, e.g. cross-entropy."
>
> To our knowledge, cross-entropy is a fairly standard choice for training a classifier to maximize accuracy when the classifier must be differentiable.
>
> -"In the related work section, specifically Incorporating IDK, it would be good to discuss the work of Wegkamp et al., including the paper http://www.jmlr.org/papers/volume9/bartlett08a/bartlett08a.pdf and its loss function. Also, the work of https://doi.org/10.1145/2700832 would be nice to briefly discuss. Also in the related work section, specifically AI Safety, it would be good to discuss the work of Varshney and Alemzadeh (https://doi.org/10.1089/big.2016.0051) --- in particular the 'safety reserves' and 'safe fail' sections which specifically address reject options and fairness respectively."
>
> We have included discussion of these papers in our related work section.

---

### Official Review · AnonReviewer1 · 2017-11-29
**This paper proposes a method to address a highly important problem in Machine learning (fairness) as regularised cost (which it not so novel)**

**Rating:** 5
**Confidence:** 3

**Review:**

The proposed method is a classifier that is fair and works in collaboration with an unfair (but presumably accurate model). The novel classifier is the result of the optimisation of a loss function (composed of a part similar to a logistic regression model and a part being the disparate impact). Hence, it can be interpreted as a logistic loss with a fairness regularisation.

The results are promising and the applications are very important for the acceptance of ML approaches in the society. However, I believe that the model could be made more general (than a fairness regularized logistic loss) and its theoretical properties studied.
Finally, this paper used uncommon vocabulary (for the machine learning community) and it make is difficult to follow sometimes (for example, the use of a Decision-Maker entity).

When reading the submitted paper, it was unclear (until section 6) how deferring could help fairness. Hence, the structure of the paper could maybe be improved by introducing the cost function earlier in the manuscript (as a fairness regularised loss).

To conclude, although the application is of high interest and the numerical results encouraging, the methodological approach does not seem to be very novel.

Minor comment :
- The list of authors of the reference “Machine bias : theres software…” apperars incorrectly (some comma may be missing in the .bib file) and there is a small typo in the title.

Possible extensions :
- The proposed fairness aware loss could be made more general (and not only in the case of a logistic model)
- It could also be generalised to a mixture of biased classifier (more than on DM).

Edited :
As noted by a fellow reviewer, the paper is a bit out of the scope of ICLR and may be more in line with other ML conferences.

---

> ### Author Response · Authors · 2017-12-21
> **Response to Reviewer 1**
>
> Thank you for the comments. We're glad you agree this is an important and promising research direction. A couple of quick notes in response:
>
> -"However, I believe that the model could be made more general (than a fairness regularized logistic loss) and its theoretical properties studied."
>
> In section 5.2, in which we describe how the "learning to defer" framework can be used in a Bayesian weight uncertainty setting. Our framework for learning to defer is extremely general - the models included do not have to be regularized logistic losses; they can be any type of model trained on any supervised loss, as long as they have some type of uncertainty/deferral output. In fact, it is the generality of the framework that makes theoretical analysis difficult.
>
> -"this paper used uncommon vocabulary (for the machine learning community) and it make is difficult to follow sometimes (for example,  the use of a Decision-Maker entity)."
>
> We are sorry the vocabulary was hard to follow. Figure 1 contains a diagram of a typical system containing an IDK model, and we describe the role of the DM in section 3; however, we have re-written the introduction and other sections to clarify the terms.
>
> -"it was unclear (until section 6) how deferring could help fairness. Hence, the structure of the paper could maybe be improved by introducing the cost function earlier in the manuscript (as a fairness regularised loss)."
>
> Thanks for the suggestion. We describe in section 3 how deferring can help accuracy, but we do not describe how deferring can help fairness. We have extended the example in section 3 to rectify this.
>
> -"The list of authors of the reference ???Machine bias : theres software?????? apperars incorrectly (some comma may be missing in the .bib file) and there is a small typo in the title."
>
> Thank you, we have corrected these typos.
>
> -"The proposed fairness aware loss could be made more general (and not only in the case of a logistic model)"
>
> Any of the fair learning and rejection learning methods can be used within the "learning to defer" framework; in this paper we demonstrated two possible options.
>
> -"It could also be generalised to a mixture of biased classifier (more than on DM)."
>
> We agree that this would be an interesting future direction of research.

---

### Official Review · AnonReviewer2 · 2017-11-30
**Conceptually very interesting, but lacks technical novelty**

**Rating:** 6
**Confidence:** 3

**Review:**

Strengths:
1. This paper proposes a novel framework for ensuring fairness in the classification pipeline. To this end, this work explores models that learn to defer.
2. The work is conceptually very interesting. The idea of learning to defer (as proposed in the paper) as a means to fairness is not only novel but also quite apt.
3. Experimental results demonstrate that the proposed learning strategy can not only increase predictive accuracy but also reduce bias in decisions.

Weaknesses:
1. While this work is conceptually quite novel and interesting, the technical novelty and contributions seem fairly minimal.
2. The proposed formulations are essentailly regularized variants of fairly standard classification models and the optimization also relies upon standard search procedures.
3. Experimental analysis on deferring to a biased decision maker (Section 7.3) is rather limited.

Summary: This paper proposes a novel framework for ensuring fairness in the classification pipeline. More specifically, the paper outlines a strategy called learn to defer which enables the design of predictive models which not only classify accurately and fairly but also defer if necessary. Deferring a decision is used as a mechanism to ensure both fairness and accuracy. Furthermore, the authors consider two variants depending on if the model has some information about the decision maker or not. Experimental results on real world datasets demonstrate the effectiveness of the proposed approach in building an end to end pipeline that ensures accuracy and fairness.

Novelty: The main novelty of this work stems from the idea of introducing learning to defer mechanisms in the context of fairness. While the ideas of learning to defer have already been studied in the context of classification models, this is the first contribution which leverages learning to defer strategy as a means to achieve fairness. However, beyond this conceptual novelty, the work does not demonstrate a lot of technical novelty or depth. The objective functions proposed are simple extensions of work done by Zafar et. al. (WWW, AISTATS 2017). The optimization procedures being used are also fairly standard. Furthermore, the authors do not carry out any rigorous theoretical analysis either.

Other detailed comments:
1. I would strongly encourage the authors to carry out a more in-depth theoretical analysis of the proposed framework (Refer to "Provably Fair Representations" McNamara et. al. 2017)
2. Experimental evaluation can also be strengthened. More specifically, analysis in Section 7.3 can be made more thorough. Instead of just sticking to one scenario where the decision maker is extremely biased (how are you quantifying this?), how about plotting a graph where x axis denotes the extent of bias in decision-maker's judgments and y-axis captures the model performance?
3. Overall, the paper is quite well written and is well motivated. There are however some typos and incorrect figure refernces (e.g., Section 7.2 first line, Figure 7.2, there is no such figure).

---

> ### Author Response · Authors · 2017-12-21
> **Response to Reviewer 2**
>
> Thank you for the comments. We're happy you found the work interesting, and we appreciate the constructive criticism. We'd like to clarify a couple of points regarding our contribution.
>
> -"While the ideas of learning to defer have already been studied in the context of classification models, this is the first contribution which leverages learning to defer strategy as a means to achieve fairness."
>
> It is true that we are the first to consider deferring as a means to achieve fairness. However, it is important not to confuse "learning to punt" with "learning to defer". As we mention in our Related Work section, what we call "learning to punt" has been studied extensively in the context of classification models (also known as rejection learning, KWIK learning, learning to abstain, IDK models, among others).  The other main contribution of our paper is the "learning to defer" framework, which extends the concept of learning to punt to consider other decision makers in the pipeline. As we motivate in sections 1, 3, and 6, and demonstrate in section 7, learning to defer can greatly enhance the real-world performance of models which punt - this performance can be measured as accuracy, fairness, or any other supervised loss function.
>
> -"the work does not demonstrate a lot of technical novelty or  depth. The objective functions proposed are simple extensions of work done by Zafar et. al. (WWW, AISTATS 2017)"
>
> The objective function for learning to defer is novel, and unrelated to Zafar et al (https://people.mpi-sws.org/~mzafar/papers/disparate_impact.pdf and https://people.mpi sws.org/~mzafar/papers/disparate_mistreatment.pdf). The fair punting objectives described in equation 4 and 5 use a similar regularization approach - we mention in our Fair Classification section that this approach is not novel, citing Kamashima et al. and Bechavod & Ligett. We could have used any fair supervised learning algorithm in place of this, and the results would hold.
>
> -"analysis in Section 7.3 can be made more thorough. Instead of just sticking to one scenario where the decision maker is extremely biased (how are you quantifying this?), how about plotting a graph where x axis denotes the extent of bias in decision-maker's judgments and y-axis captures the model performance?"
>
> As we state in our caption of figure 5, we train a biased DM by setting alpha (the coefficient on the fairness regularization term) to -0.1, thereby encouraging solutions with higher disparate impact.
>
> We cannot plot model performance on y-axis due to the complexity of the evaluative metric - assessing the accuracy-fairness tradeoff requires a two-dimensional Pareto-front style visualization. We have produced the same graph for several values of alpha, and did not think it was too illuminating; but we could include it in the Appendix.
>
> -"There are however some typos and incorrect figure references (e.g., Section 7.2 first line, Figure 7.2, there is no such figure)."
>
> Thank you, we have corrected the figure reference.

---

### Decision · Program_Chairs · 2018-01-29
**ICLR 2018 Conference Acceptance Decision**

**Decision:**

Invite to Workshop Track

**Comment:**

This work is proposing an approach for ensuring classification fairness through models that encapsulate deferment criteria. On the positive side, the paper provides ideas which are conceptually interesting and novel. On the other hand, the reviewers find the technical contribution to be limited and, in some cases, challenge the practicality of the method (e.g. requirement for second set of training samples). After extensive post-rebuttal discussion, the consensus is that the above issues make the paper fall below the threshold for acceptance – even if the “out-of-scope” issue is not taken into account.